# Lim Domain Binding 3 (Ldb3) Identified as a Potential Marker of Cardiac Extracellular Vesicles

**DOI:** 10.3390/ijms23137374

**Published:** 2022-07-01

**Authors:** Fadi Abou Zeid, Henri Charrier, Olivia Beseme, Jean-Baptiste Michel, Paul Mulder, Philippe Amouyel, Florence Pinet, Annie Turkieh

**Affiliations:** 1U1167-RID-AGE-Facteurs de Risque et Déterminants Moléculaires des Maladies Liées au Vieillissement, Institut Pasteur de Lille, Université de Lille, Inserm, CHU Lille, 59000 Lille, France; fadi.abouzeid@pasteur-lille.fr (F.A.Z.); henri.charrier@genoscreen.fr (H.C.); olivia.beseme@pasteur-lille.fr (O.B.); philippe.amouyel@pasteur-lille.fr (P.A.); 2U1116-DCAC, Université de Lorraine, Inserm, 54000 Nancy, France; jean-baptiste.michel@inserm.fr; 3Inserm U1096, UNIROUEN, Normandie University, 76000 Rouen, France; paul.mulder@univ-rouen.fr

**Keywords:** Extracellular vesicles, proteomic analysis, LIM Domain Binding 3, cardiac EV marker, ischemic heart failure, EVtrap

## Abstract

Extracellular vesicles (EVs) are considered as transporters of biomarkers for the diagnosis of cardiac diseases, playing an important role in cell-to-cell communication during physiological and pathological processes. However, specific markers for the isolation and analysis of cardiac EVs are missing, imposing limitation on understanding their function in heart tissue. For this, we performed multiple proteomic approaches to compare EVs isolated from neonate rat cardiomyocytes and cardiac fibroblasts by ultracentrifugation, as well as EVs isolated from minced cardiac tissue and plasma by EVtrap. We identified Ldb3, a cytoskeletal protein which is essential in maintaining Z-disc structural integrity, as enriched in cardiac EVs. This result was validated using different EV isolation techniques showing Ldb3 in both large and small EVs. In parallel, we showed that Ldb3 is almost exclusively detected in the neonate rat heart when compared to other tissues, and specifically in cardiomyocytes compared to cardiac fibroblasts. Furthermore, Ldb3 levels, specifically higher molecular weight isoforms, were decreased in the left ventricle of ischemic heart failure patients compared to control groups, but not in the corresponding EVs. Our results suggest that Ldb3 could be a potential cardiomyocytes derived-EV marker and could be useful to identify cardiac EVs in physiological and pathological conditions.

## 1. Introduction

Extracellular vesicles (EVs) are lipid bilayer-bound particles secreted by almost all cell types into the extracellular space [1]. Based on their biogenesis, size, and composition, different subtypes of EVs exist, such as exosomes, microvesicles, and apoptotic bodies [2]. EVs carry molecular cargo such as proteins, lipids, and RNA, playing a role in cell-to-cell communication during physiological and pathological processes [3,4]. The content of EVs varies depending on the type and functional state of cells that produce them [5,6]. Furthermore, the double-layered membrane protects the transported molecules from degradation in biological fluids, increasing their stability. For these reasons, EVs are considered as a source of biomarkers and could be used to diagnose pathologies, including cardiac diseases [7,8,9,10,11].

In the heart, different types of cells, including cardiomyocytes, fibroblasts, and endothelial cells, communicate via EVs to maintain cardiac homeostasis [12]. It was shown that cardiac cells can secrete EVs whose number, size, and/or content differ with different pathological status [13,14,15,16]. Borosch et al. showed that preconditioning with hypoxia and isoflurane alters the protein levels of cardiomyocyte- and fibroblast-derived EVs differently [17]. However, the same stimuli did not have the same effect on EV composition depending on their producing cells. Furthermore, EVs secreted by different cardiac cells have been shown to regulate cardiac remodeling (hypertrophy, fibrosis, inflammation) in an autocrine or paracrine fashion [4,16,18,19,20]. All these data showed the interest in deeply characterizing the cardiac vesicles to better understand their role in heart disease.

EV characterization is a complex process due to their high heterogeneity and to the currently used protocols that cannot guarantee EV purity. Most studies were performed by using cell lines or body fluids. Studies on tissue-derived EVs were increased, but their number is still limited. Proteomic analysis has been used to characterize EVs from cardiac tissue [6,21]. However, variability in protein identification is observed depending on how samples are collected and which method is used to isolate EVs. Consequently, no cardiac-specific markers are fully characterized. Furthermore, previous in vitro proteomic analyses were performed on EVs derived from the same cardiac cell type under different stress conditions [13,14,17]. To our knowledge, no studies have been performed comparing EVs derived from different cardiac cell types with the goal of identifying cell-specific markers.

The aim of this study was to identify specific protein markers that both distinguish cardiac EVs from non-cardiac EVs in the circulation, as well as distinguish the different cardiac cell-derived EVs from each other in order to better understand their role. We identified and validated Ldb3, a cytoskeletal protein which is essential in maintaining Z-disc structural integrity, as a potential cardiomyocyte EV-specific marker via different EV isolation techniques.

## 2. Results

### 2.1. Comparative Proteomic Profiling of Cardiac Cell-Derived EVs Isolated by Differential Ultracentrifugation

In order to identify specific markers of cardiac cell-derived EVs, we isolated, by differential ultracentrifugation, large (lEVs) and small EVs (sEVs) from conditioned culture media of neonate cardiac myocytes (NCM) and fibroblasts (NCF) as well as cardiomyoblasts H9c2 (Figure 1A). Nanoparticle tracking analysis (NTA) (Figure 1B and Appendix A) showed that NCFs produced the highest amount of EVs (sEVs: 8.9E6 particles/cell; lEVs: 8.5E5 particles/cell), compared to H9c2 cells (sEVs: 1.2E5 particles/cell; lEVs: 6E4 particles/cell) and NCM (sEVs: 4.4E4 particles/cell; lEVs: 6E3 particles/cell); sEV production was also consistently higher than lEVs in all cell types. Concerning size distribution, NCF EVs displayed a wide difference in size between sEVs and lEVs (sEVs: 120.4 nm/lEVs: 164.3 nm), which was less pronounced in NCM EVs (sEVs: 97.3 nm/lEVs: 107.9 nm) and H9c2 EVs (sEVs: 130 ± 12 nm/lEVs: 145 ± 13 nm).

Due to the low abundance of NCM derived EVs, only NCF- and H9c2-derived sEVs were further analyzed for proteomic characterization. Western blot analysis shows increased protein levels of classical exosomal marker CD63 [22] in H9c2- and NCF-derived sEVs (Figure 1C), validating the presence of exosomes in sEVs. The absence of CD9 in these sEVs suggests a specific protein profile of cardiac EVs.

We then performed gel separation and mass spectrometry of proteins revealed by Coomassie blue staining to determine the protein composition of NCF- and H9c2-derived sEVs (Figure 1D). We identified 63 proteins from NCF and 8 proteins from H9c2, including 5 common proteins (such as Col1a2, Fn1, Mfge8, Actb, and Actg1) (Figure 1E, Appendix A). Gene ontology (GO) analysis (Figure 1F) for cellular component revealed 24 cytoskeletal proteins (such as actin and tubulin), 12 ribonucleoproteins (such as Rps2/4x/8/9 and Rpl7a/11/15), 6 extracellular matrix proteins (such as Col1a1/2 and Fn1), and 6 DNA-binding proteins (such as histones H4m/2a1e/3-3b/2aj and Morf4l2). Seventeen proteins were associated with positive regulation of protein expression (such as Hspa8, Wnt5a, and Eef1), and 15 with regulation of the apoptotic process (such as Hsp90aa1/a5, Grk2, Rack1, and Tcp1).

### 2.2. Comparative Proteomic Profiling of Cardiac and Plasma-Derived EVs Isolated by EVtrap

To go deeper in understanding the cardiac EV proteome, a more sensitive approach was designed. EV total recovery and purification (EVtrap) was used to purify EVs from the extracellular environment surrounding cardiac cells (Figure 2A).

A total of 3187 EV proteins were detected (Appendix A), of which 2629 cardiac proteins and 1905 plasma proteins were analyzed (detected in at least 3 out of 4 samples in each group), with 1775 proteins in common (Appendix A, Appendix A). Among identified proteins, we detected most of the proteins (41 out of 66) previously detected in sEVs derived from cardiac cells.

Cardiac EV proteins were compared with the 100 most frequently identified EV proteins according to EV databases ExoCarta and Vesiclepedia. Respectively, 91 and 86 of these proteins were detected in cardiac EVs (Figure 2B, Appendix A).

GO analysis (Figure 2C, Appendix A) revealed the presence of typical exosome markers (such as Sdcbp/syntenin–1, Pdcd6ip/alix, tetraspanins CD81/9, Hspa4/8, Lamp2, and Tsg101). Cardiac vesicles were also enriched in membrane proteins (such as annexins, cadherins, cavins, caveolins, clusters of differentiation, and integrins), endosomal proteins (such as Chmp, Rab, EH-domain containing, vacuolar protein sorting-associated, and sorting nexin proteins), focal adhesion proteins (such as LIM domain-containing proteins), and sarcomeric proteins.

Abundance of cardiac and plasma EV proteins were then compared showing 1033 cardiac EV-enriched proteins, while 180 proteins were plasma EV-enriched (Figure 2D, Appendix A). Compared to plasma EVs, we showed that Ldb3 is the most significantly abundant protein (×160, *p* = 0.03) in cardiac EVs (Figure 2D–F) while the previously reported cardiac marker Cryab was less enriched (×40, *p* = 0.03) (Figure 2D,F).

### 2.3. Ldb3 as a Potential Cardiac EV Marker

To validate that Ldb3 is a cardiac-specific marker, its expression was evaluated in various neonate rat tissues (lung, liver, kidney, heart, plasma, leg muscle, brain, and skin). Ldb3 protein level was higher in heart tissue, confirming its relative cardiac specificity, while low amounts were detected in the brain (Figure 2G). Cryab was also found to be cardiac-enriched, with low amounts in leg muscle tissue. Additionally, within cardiac tissue Ldb3 expression was cardiomyocyte-specific compared to cardiac fibroblasts, suggesting that Ldb3 could be a marker of cardiomyocytes-derived EVs (Figure 2G).

In order to further confirm Ldb3 as a cardiac EV marker, isolation of neonate rat cardiac EVs was performed using different approaches: ultracentrifugation (UC), polyethylene glycol precipitation (PEG), or a mixed technique combining the two (PEG + UC) (Figure 3A).

The different EV populations were analyzed by Western blotting. We found that Ldb3, Syntenin–1, an exosomal marker, and Cryab, were abundant in EV samples (Figure 3B, left) compared to ventricular tissue. Syntenin–1 was most enriched in EVs obtained by PEG + UC, implying higher exosomal retention using this approach. Cryab was detected similarly in the 10K pellet, UC, PEG, and PEG + UC, indicating that it may be loaded into both lEVs and sEVs, with a higher concentration being loaded into sEVs. Ldb3 also appeared to be enriched in both lEVs and sEVs compared to heart tissue. Ldb3 was most abundant in sEVs that were isolated using PEG + UC compared to UC and PEG alone.

These techniques were applied to obtain EVs from ventricles and plasma in order to confirm that Ldb3 is cardiac EV-specific (Figure 3B, right). Both Ldb3 and syntenin–1 enrichment were detected in PEG + UC compared to the other techniques, and Ldb3 was only observed in cardiac EV samples, validating the EVtrap result. It should be noted that the expression profile of Ldb3 differed between samples, as two isoforms appeared in different EV samples. The higher MW isoform was enriched in UC-EVs, while the lower isoform appeared with PEG. Both were present in PEG + UC, indicating a possible influence of Ldb3 isoform on which EV subpopulation it will be loaded into.

### 2.4. Size Exclusion Chromatography Validated Ldb3 Loading into EVs

Size exclusion chromatography (SEC) was used to separate cardiac EVs from non-vesicular components (NV) (Figure 4A).

The non-purified sample (input) was loaded into a SEC column and 30 fractions were analyzed. NTA and protein concentration measurement for each fraction shows that particle concentration was the highest on fraction 7, progressively reducing until the final fractions, while protein concentration increased (Figure 4B). These results, in addition to TEM imaging of pooled fractions, confirm that EVs were mainly eluted in the fractions 7–12, with NV eluting mostly after fraction 12. SEC fraction 20 was taken as NV for Western blot comparison with EV fractions obtained by UC and the original input sample (Figure 4C). Ldb3 was enriched in EV fractions compared to the NV fraction. Conversely, albumin appeared to be enriched in the NV fraction, indicating successful separation of EVs from NV. This confirms that Ldb3 is loaded into EVs and is not co-pelleting with non-vesicular components during EV isolation.

### 2.5. Ldb3 Levels Are Increased in Left Ventricle of Heart Failure Rats and Patients with Ischemic Cardiomyopathy but Not in Cardiac EVs

We previously observed an alteration of sarcomere structure in the left ventricle (LV) of rats with coronary ligated heart failure (HF) [23]. We were therefore interested in studying cardiac Ldb3 expression during HF following myocardial infarction. We found that due to alternative splicing, many Ldb3 isoforms were observed in rat heart, and the profile of this protein differs in neonatal and adult rats (Appendix A). Moreover, levels of Ldb3 (high isoforms) decreased in HF rats compared to control, while low Ldb3 isoform (~32 kDa) remained unchanged. This result was validated in LV of patients with ischemic cardiomyopathy who underwent a cardiac transplantation (Figure 5A).

Next, we verified if Ldb3 levels change in cardiac EVs during HF. Cardiac tissue from 6 control and 6 ischemic HF patients were minced and digested with collagenase II, then UC was used to isolate EVs. As observed by NTA analysis (Figure 5B), we isolated more lEVs than sEVs, and the amount and the size distribution profile of these EVs was similar between control and HF patients. Contrary to what was observed in LV, only one Ldb3 isoform was detected (high, ~78 kDa), and its level was not changed between EVs of control and HF samples (Figure 5C).

## 3. Discussion

In this study, our aim was to identify specific EV markers of cardiac EVs allowing us to distinguish them from non-cardiac EVs in the circulation. Here, we identified Ldb3 as a potential cardiomyocyte-derived EV marker.

EVs are considered as transporters of biomarkers for the diagnosis of cardiac diseases, playing an important role in cell-to-cell communication during physiological and pathological processes, hence the interest in deeply characterizing these vesicles to better understand their role in heart disease.

EV characterization is a complex process due to their high heterogeneity and the lack of methods to fully separate different EV subpopulations [24,25]. Furthermore, currently used protocols cannot guarantee EV purity [26,27]. Previous studies were performed to isolate and characterize EVs from cardiac tissue from minced or perfused cardiac tissue with or without enzymatic digestion [6,18,21,28]. Proteomic studies widely used UC with or without sucrose gradient to isolate EVs. Differences in protein identification were observed depending on how samples were collected and which method was used to isolate them. The protein profile for EVs isolated from different rat tissues by UC followed by sucrose gradient was compared and identified Caveolin-3 (Cav3) as a marker of cardiac EVs, but this microvesicle marker [29] could be expressed by other tissues. In order to find specific cardiac EV markers, we performed a comparative proteomic analysis on EVs isolated by EVtrap from neonate rat plasma and cardiac tissue. EVtrap is a novel, fast, reproducible, and high-recovery EV isolation method with relatively low contamination levels [30]. It was used to isolate EVs from plasma and urine for phosphoproteomic analysis [30,31]. This technique was used for the first time to isolate EVs from cardiac tissue. Compared to previous studies using UC, little amount of tissue is required. Both lEVs and sEVs were collected. In addition to the 2 cardiac EVs markers previously characterized, Cav3 and Cryab, we identified Ldb3 as most significantly enriched in cardiac EV compared to plasma EVs. Ldb3 was detected in both cardiac lEVs and sEVs isolated by other techniques which are cheaper and more available than EVtrap. We confirmed that Ldb3 is more enriched in cardiac sEVs compared to plasma EVs. Comparing the expression of Ldb3 and albumin in unfiltered EVs, lEVs and sEVs obtained by UC, and SEC non-vesicular fraction confirmed that Ldb3 is loaded into EVs and is not a protein contaminant. We then validated the presence of Ldb3 in human cardiac EVs isolated by UC. All this data suggests Ldb3 as a potential cardiac- EVs marker.

Furthermore, we have demonstrated that Ldb3 is almost exclusively expressed in the heart compared to other neonate rat tissue, and specifically in cardiomyocytes compared to cardiac fibroblasts, suggesting Ldb3 as a novel cardiomyocyte-derived EV marker. Similarly to Cav3 and Cryab, Ldb3 is expressed in skeletal muscle. Ldb3, also known as Cypher (in murines) or ZASP (in humans), is a PDZ and LIM domain-containing protein that localizes to the Z-disc in striated skeletal and cardiac muscle cells [32]. Due to alternative splicing, several forms of Ldb3 exist which may be exclusively skeletal or cardiac, or common isoforms for both tissues. It has been shown that the Ldb3 isoform possessing exon 4 are predominantly expressed in cardiac tissue, whereas the isoforms possessing exon 6 are more prominent in skeletal muscle [33]. In our study, we observed that several Ldb3 isoforms exist in heart tissue, while only one isoform was secreted in EVs. A possible way to identify whether the EV-secreted isoform is cardiac-specific is by using antibodies that bind specifically to the cardiac exon (exon 4). Notably, we observed that cardiac Ldb3 isoforms differ between neonate and adult rats, implying age-dependent changes in its splicing and function.

Ldb3 is part of the protein complex that maintains the structural integrity of sarcomeres during muscle contraction by binding to alpha-actinin and myotilin via its PDZ domain and plays a role in signaling through its interactions with PKC via its LIM domain [34,35]. It was shown that deletion of Ldb3 in a mouse model resulted in neonatal lethality associated with Z-disc disorganization and fragmentation in both skeletal and cardiac muscle cells [36,37]. Furthermore, Ldb3 variants have been observed to be associated with myopathy in human patients [38,39]. In this study, we showed that the expression of high MW Ldb3 isoforms decreased in LV of HF patients with ischemic cardiomyopathy, as well as in LV of HF rats in which we have previously observed LV structural alterations, desmin aggregate accumulation, and sarcomere disruption [23,40]. However, no difference on Ldb3 levels were observed in lEVs and sEVs isolated from HF patients compared to control. These results showed that intraventricular and not EV-Ldb3 levels are associated with HF.

In summary, we showed the presence of Ldb3 in cardiac EVs by using different sample preparation methods, fresh or frozen tissue, minced with or without enzymatic digestion, and different isolation techniques. We suggest that Ldb3 can act as a potential marker of cardiac EVs specifically derived from cardiomyocytes. It should be noted that despite using several EV isolation techniques, it is currently not possible to completely eliminate contamination of EV samples with soluble proteins, lipoprotein aggregates, intracellular vesicles, or organelle vesicles produced by cell damage. Complementary experiments can be performed to confirm Ldb3 loading into EVs and its presence in released EVs. Ldb3 localization in EVs is also unclear. Future experiments may validate Ldb3 presence inside or at the surface of EVs. It is likely loaded inside EVs along other EV cargo proteins, but may also associate with integrin proteins on EV membranes, allowing for its potential use as an EV immunocapture target.

To conclude, this is the first study showing the presence of Ldb3 in cardiac EVs and suggests that Ldb3 could be used to identify these EVs in physiological and pathological conditions.

## 4. Materials and Methods

### 4.1. Cellular Models

#### 4.1.1. Primary Cultures of Neonate Rat Cardiomyocytes (NCM) and Cardiac Fibroblasts (NCF)

NCM and NCF were isolated from heart ventricles of one- to three-day old Wistar rats (Janvier Labs, Le Genest-Saint-Isle, France) as previously described [41]. Briefly, minced ventricular tissue was digested with type II collagenase (LS004174, Worthington, Lakewood, NJ, USA) and pancreatin (P3292, Sigma-Aldrich, St. Louis, MO, USA), and dissociated cells were pelleted by centrifugation at 800× *g*. NCMs were separated from NCFs via discontinuous Percoll gradient (bottom 58.5%, top 40.5% [*v*/*v*], P4937, Sigma-Aldrich, St. Louis, MO, USA) for 30 min at 1600× *g*. NCMs were seeded at 2.5 × 10^6^ cells per 100 mm collagen-coated culture dish and were cultured in a 4:1 mixture of DMEM (D1152, Sigma Aldrich, St. Louis, MO, USA) and Medium 199 (M4530 Sigma-Aldrich, St. Louis, MO, USA), supplemented with 10% horse serum (HS) (16050122, Thermo Fisher Scientific, Waltham, MA, USA), 5% fetal calf serum (FCS) (30-2020, ATCC, Manassas, VA, USA), and 1% penicillin-streptomycin (PS) (15140-122, Gibco, Waltham, MA, USA) for 5 days at 37 °C under 5% CO_2_ atmosphere. NCFs were seeded at 2 × 10^6^ cells per 100 mm culture dish and were cultured in DMEM (31966-02, Gibco, Waltham, MA, USA), 10% FCS, and 1% PS for 48 h at 37 °C under 5% CO_2_ atmosphere. NCF at passage 1 were used for experiments.

#### 4.1.2. Rat Cardiac Myoblasts

H9c2 cells (CRL-1446, ATCC, Manassas, VA, USA) were cultured in similar conditions as NCFs and used at passages 8 to 15 for experiments.

### 4.2. Animal Model

All animal experiments were performed according to the Guide for the Care and Use of Laboratory Animals published by the US National Institutes of Health (NIH publication NO1-OD-4-2-139, revised in 2011). MI was induced in 10-week-old male Wistar rats (Janvier, Le Genest St isle, France) by ligation of the left anterior descending coronary artery and animals were sacrificed 2 months post-MI as previously described [42]. All tissues were stored at −80 °C before use.

### 4.3. Human Cardiac Biopsies

Human heart tissue was obtained from the cardiovascular biobank of Bichat Hospital in Paris (BB-0033-00029, coordinator Dr JB Michel) with approval by the Inserm Institutional Review Board. Explanted heart tissues were obtained from patients undergoing heart transplantation for end-stage ischemic heart failure and from patients who died of non-cardiac causes. Samples were quick-frozen and stored at −80 °C. All materials from patients were recovered as surgical waste with informed consent of the donors and with approval of the local ethical boards (“Centre Hospitalier et Universitaire de Lille”, Lille, according to the Declaration of Helsinki).

### 4.4. EV Isolation

#### 4.4.1. Sample Preparation

One- to three-day old rats were sacrificed using decapitation. Mixed venous and arterial blood from decapitated rats was collected into EDTA tubes (16.444.100, Sarstedt, Nümbrecht, Germany) and was centrifuged at 2000× *g* for 5 min without brake to obtain plasma (100 µL/rat). Hearts were quickly excised and placed in ice-cold conservation buffer ADS (6.8 g/L NaCl, 1 g/L glucose, 0.12 g/L NaH_2_PO_4_, 0.4 g/L KCl, 0.1 g/L MgSO_4_, 4.76 g/L HEPES, pH 7.4) [18]. Atria and surrounding tissue were removed. Ventricles were washed then minced in ice-cold conservation buffer (100 µL/ventricle). Adult tissue samples were prepared similarly, with the addition of a digestion step after mincing. Minced tissue (~200 mg) was incubated in 6-mL of 1 mg/mL type II collagenase (LS004174, Worthington, Lakewood, NJ, USA) at 37 °C for 30 min with agitation. To isolate EVs secreted by cells, NCM, NCF and H9c2 were incubated in a 50% reduced volume of serum-free culture medium for 96 h. EV-containing conservation buffer, plasma, and conditioned culture media were centrifuged at low speed (400 to 3000× *g*) to remove debris and dead cells. The supernatant containing unpurified EVs was used for further analyses. Unless stated otherwise, all manipulations were performed at 4 °C.

#### 4.4.2. Differential Ultracentrifugation (UC)

Unpurified EV solution was further ultracentrifuged at 4 °C at medium speed (10,000 to 20,000× *g*) for 70 min to pellet large EVs, then at high speed (100,000 to 164,000× *g*) for 70 min to pellet small EVs. Rotors used were either Beckman 50.2Ti or SW-32.1 (337901/369651 Beckman Coulter France, Villepinte, France).

#### 4.4.3. Polyethylene Glycol (PEG) Precipitation

After medium speed ultracentrifugation, supernatants were incubated with 10% PEG 6000 (P0903, TCI Chemicals, Tokyo, Japan) overnight at 4 °C. The white precipitate containing EVs formed was pelleted by a low-speed centrifugation (3000× *g*) for 15 min at 4 °C. For PEG precipitation followed by UC (PEG + UC), PEG pellet was resuspended in filtered phosphate buffered saline (PBS) and ultracentrifuged at high speed (100,000 to 164,000× *g*) for 70 min at 4 °C.

#### 4.4.4. Size Exclusion Chromatography (SEC)

Unpurified EV solution from ventricles (500 µL) were analyzed through qEV original size exclusion columns containing resins of 70 nm pore size (Izon Science, Christchurch, New Zealand), pre-washed at room temperature with filtered and degasified PBS. Fractions of 500 µL were collected with a constant flow of PBS, then stored at −80 °C until use.

#### 4.4.5. EV Total Recovery and Purification (EVtrap)

Unpurified EVs from plasma and minced ventricles were captured and processed using magnetic EVtrap beads as previously described [30]. Briefly, unpurified EV solution from ventricles (3000× *g* supernatant) was filtered using a 100,000 Da filter (Amicon^®^ Ultra-15 Centrifugal Filter Unit, UFC9100, Millipore, Burlington, MA, USA) allowing the removal of free molecules with small molecular mass. Samples were then washed and diluted 1:10 with PBS (1X). In order to isolate EVs, magnetic EVtrap beads were added to the filtered solution at 1:10 *v*/*v* ratio, followed by shaking for 1 hr. The beads were then captured using a magnetic separator rack, and the supernatant was removed. Beads were washed using PBS.

### 4.5. Nanoparticle Tracking Analysis (NTA)

Samples were analyzed using a NanoSight NS300 instrument (Malvern Panalytical, Malvern, UK) according to the manufacturer’s software manual (NanoSight NS300 User Manual, MAN0541-01-EN-00, 2017). Samples were diluted in filtered PBS (1X), and videos of 1 min were performed with the camera level setting at 14 and detection threshold at 4. Videos were analyzed with Nanosight NTA software version 3.2 build 3.1.46 (Malvern Panalytical, Malvern, UK). To compare the concentration of particles detected from different samples, values were normalized to cell number.

### 4.6. Transmission Electron Microscopy (TEM)

Five µL from each sample were deposited on a 400-mesh copper grid covered with a Formvar-carbon film for 5 min at RT. The grid was dried, and 10 µL of phosphotungstic acid (19500, EMS, Hatfield, PA, USA) was added on the grid for 5 min at RT to negatively stain the biological structures. The grid was dried and observed at 80 kV with an electron microscope (H7500, Hitachi, Tokyo, Japan) equipped with a digital camera (AMT, Woburn, MA, USA).

### 4.7. Western Blot

Proteins were extracted from organ tissue, cells, and EV pellets in RIPA buffer containing 50 mM Tris base, 150 mM NaCl, 100 mM sodium orthovanadate, 1% NP40, 1% SDS, 0.1% sodium desoxycholate, 1% Complete™ Protease Inhibitor Cocktail (11697498001, Roche Diagnostics, Basel, Switzerland), and 1% antiphosphatase, phosphatase inhibitor cocktail 2 and 3 (P5726 and P0044, Sigma Aldrich, St. Louis, MO, USA) as has been previously conducted [41]. Protein concentration was measured using Pierce™ BCA Protein Assay Kit (23225, Thermo Fisher Scientific, Waltham, MA, USA) or Bradford assay (#5000006, Bio-Rad Laboratories, Hercules, CA, USA) according to manufacturer instructions. Equal amounts of protein were diluted in reducing sample buffer and boiled for 10 min at 70–95 °C. Samples were loaded into Bio-Rad or NuPAGE gels (Invitrogen, Waltham, MA, USA) then migrated. Transfer was performed using Trans-Blot Turbo™ transfer system (Bio-Rad Laboratories) for 10 min at 1.3 A and 25 mV onto a 0.2 µm nitrocellulose membrane (170-4158, Bio-Rad Laboratories, Hercules, CA, USA). Membranes were blocked using 5% milk in TBS-tween solution (0.1 M Tris HCl pH 7.4, 0.1% [*v*/*v*] tween, 150 mM NaCl), then incubated with primary antibodies overnight at 4 °C (Appendix A). Incubation with HRP-conjugated secondary antibodies at a dilution of 1:5000–1:10,000 was performed the following day for 1 h at RT. Proteins were detected by incubating membranes in Clarity™ Western ECL Substrate (Bio-Rad Laboratories, Hercules, CA, USA) for 5 min. Images were acquired using ChemiDoc Imaging System (Bio-Rad Laboratories, Hercules, CA, USA).

### 4.8. Proteomics

EV proteins isolated from NCF and H9c2 separated using sodium dodecyl sulfate–polyacrylamide gel electrophoresis were fixed with 50% ethanol and 2% orthophosphoric acid for 1 h at RT. Gels were washed, then incubated in a pre-soak solution (15% ammonium sulfate, 17% ethanol, 2% orthophosphoric acid) for 20 min at RT. Gel fragments were excised and proteins were digested using trypsin (V5280, Promega, Madison, WI, USA) followed by MALDI-TOF/TOF analysis as described previously [41].

EVs captured using EVtrap were lysed for protein extraction using a phase-transfer surfactant (PTS)-aided procedure. Proteins were reduced and alkylated with 10 mM TCEP and 40 mM CAA at 95 °C for 10 min. Samples were diluted five-fold with 50 mM triethylammonium bicarbonate and digested with Lys-C (Wako, Osaka, Japan) at 1:100 [wt/wt] for 3 h at 37 °C, before overnight trypsin digestion (1:50 wt/wt) at 37 °C. LC-MS/MS analysis was performed as described previously [30].

### 4.9. Data Analysis

Bar and column graphs were generated using GraphPad Prism (v7.0.0). EVtrap protein abundance was analyzed using Perseus v1.6.5 software (https://maxquant.net/perseus/, accessed on 1 October 2019). Proteins not detected in at least 3/4 of the samples (EVtrap analysis) in each group were excluded. Protein abundances were transformed to Log2. Missing values were substituted by values from the normal distribution. Groups were compared by a two-tailed Student’s *t* test (*p* < 0.05) using the Benjamini–Hochberg correction. Differences in abundance (Log2 transformed) and *p* values were visualized on a Volcano plot by entering Perseus data into VolcaNoseR web app [43].

Gene Ontology (GO) analysis was performed on proteins identified by MALDI-TOF/TOF and LC-MS/MS using PANTHER GO [44,45,46]. Protein names were compared to the available Rattus norvegicus reference proteome (21,585 proteins; released 2020-11-01).

EVtrap proteins selected after MS identification were compared to Vesiclepedia [47] and Exocarta [48] databases using FunRich (v3.1.3) [49]. Overlap values between the 3 groups were visualized using the Eulerr R package [50].

## Figures and Tables

**Figure 1 ijms-23-07374-f001:**
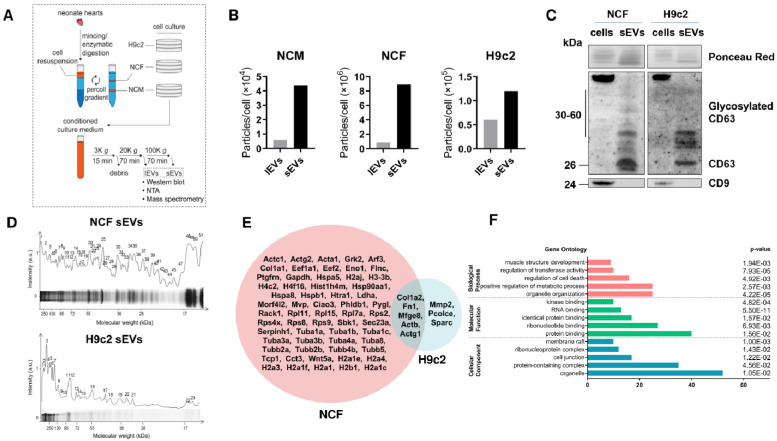
Proteomic analysis of cardiac cells-derived small EVs. (**A**) Isolation procedure of rat cardiac cells and their corresponding EVs. Neonate cardiac myocytes (NCM) and fibroblasts (NCF) isolated from ventricles of 1–3 day old Wistar rats, and cardiomyoblasts H9c2 were cultured in serum-free media for 96 h. Differential ultracentrifugation was used to isolate large (lEVs) and small EVs (sEVs) from conditioned culture medium. K: ×1000. (**B**) Nanoparticle tracking analysis (NTA) measurement of particle concentration in lEV and sEV pellets normalized to the number of producing cells in NCM, NCF, and H9c2 cells. Details are shown in Appendix A. (**C**) Ponceau Red and Western blot of cells and corresponding sEVs to detect exosomal markers CD63 and CD9. (**D**) Coomassie blue staining of gel separated NCF (75 µg) and H9c2 (25 µg) sEV proteins. Highly stained bands (labeled with a number) were used for MALDI-TOF/TOF mass spectrometry (MS). (**E**) Venn diagram representation of sEV proteins detected by MS in NCF (pink) and H9c2 (blue) sEVs. Protein abbreviations are detailed in Appendix A (**F**) GO (Gene Ontology) analysis of enriched GO terms for biological process, molecular function, and cellular component. *p* values were FDR-adjusted.

**Figure 2 ijms-23-07374-f002:**
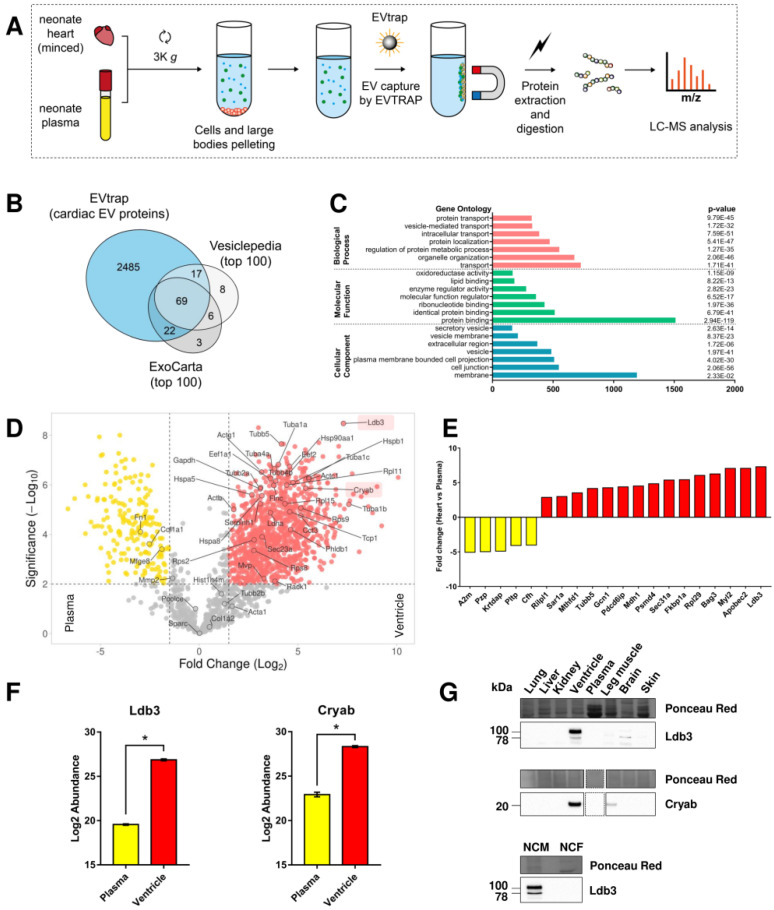
Proteomic analysis of neonatal rat heart and plasma EVs isolated by lipidic affinity. (**A**) Isolation of cardiac and plasma EVs (n = 4) using EV Total Recovery And Purification (EVtrap). Ventricles of 1–3 day old Wistar rats were minced using fine scissors before being resuspended in ice-cold ADS buffer. EVs were purified from ADS (cardiac) and plasma by binding to magnetic beads with lipidic affinity to membrane-bound particles, then lysed for analysis by liquid chromatography-mass spectrometry (LC-MS). K: ×1000. (**B**) Venn diagram of cardiac EV proteins (blue) compared to top 100 EV proteins identified in the public databases Vesiclepedia and ExoCarta. (**C**) GO analysis on the cardiac EV proteins to identify enriched GO terms. (**D**) Volcano plot comparing cardiac (red dots) and plasma (yellow dots) EV protein abundances. Proteins with *p* value < 0.01 and fold change > 1.5 were considered significant for the analysis. Only proteins detected by previous MALDI-TOF analysis, in addition to Ldb3 (LIM domain binding 3) and Cryab (Alpha-crystallin B chain), were labeled. (**E**) Bar plot representing the most significantly different (lowest *p*-value) proteins in plasma (yellow bars) and cardiac (red bars) EVs. (**F**) Bar plot of Ldb3 and Cryab abundances (log2 transformed) in plasma and heart. Error bars = SEM, * *p* < 0.05. (**G**) Quantification by Western blot of Ldb3 and Cryab expression in different tissues obtained from neonatal rat (upper panel) and in neonate cardiac myocytes (NCM) and fibroblasts (NCF) (lower panel). Equal amount of proteins were loaded for each blot as shown by Ponceau Red staining.

**Figure 3 ijms-23-07374-f003:**
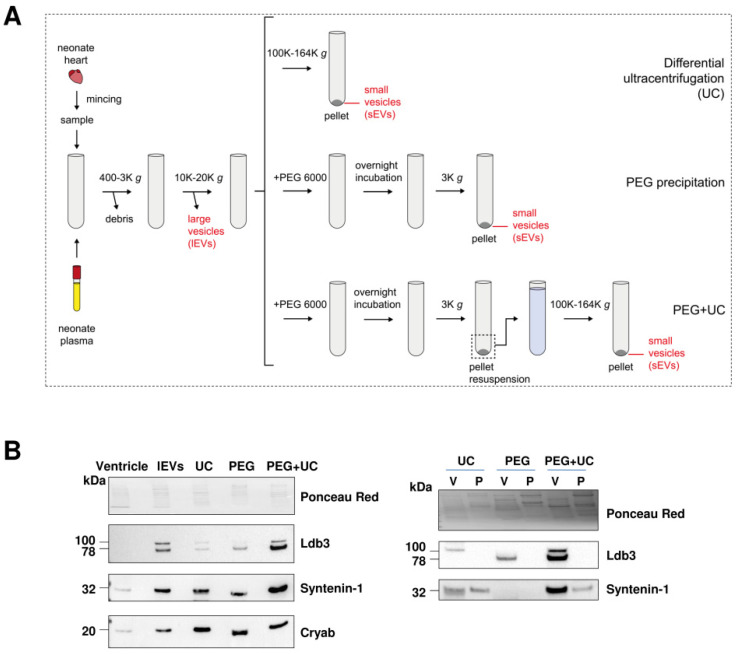
Ldb3 is enriched in cardiac EVs compared to heart tissue and plasma EVs. (**A**) Scheme of the three EV isolation techniques from neonate rat heart: ultracentrifugation (UC), polyethylene glycol precipitation without (PEG) or with UC (PEG + UC). K: ×1000. (**B**) Detection of Ldb3, syntenin–1, and Cryab by Western blot of EV populations obtained using the three isolation techniques. Equal amount of proteins were loaded for each blot as shown by Ponceau Red staining. V: ventricle; P: plasma.

**Figure 4 ijms-23-07374-f004:**
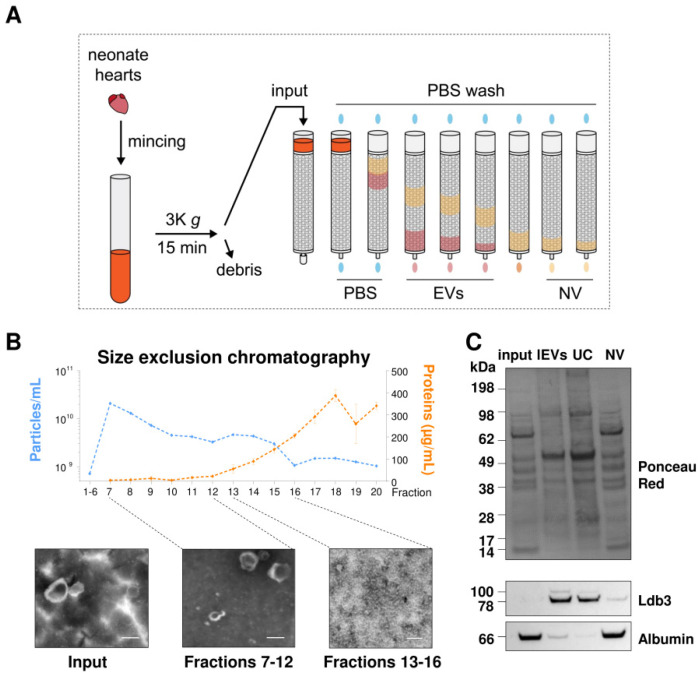
Ldb3 is a specific marker of cardiac EVs. (**A**) Protocol of size exclusion chromatography (SEC) to separate EVs from non-vesicular factors (NV). K: ×1000. (**B**) Graph representation of NTA particle count (blue) and protein concentrations (orange) for each SEC fraction. Representative transmission electron microscopy images of unpurified EVs (solution obtained after centrifugation at 3000× *g*) and different pools of SEC fractions (7–12 and 13–16) are presented below. Scale bars = 150 nm. (**C**) Quantification of Ldb3 and albumin by Western blot of EVs obtained by differential ultracentrifugation (UC) and in SEC fraction 20 (NV).

**Figure 5 ijms-23-07374-f005:**
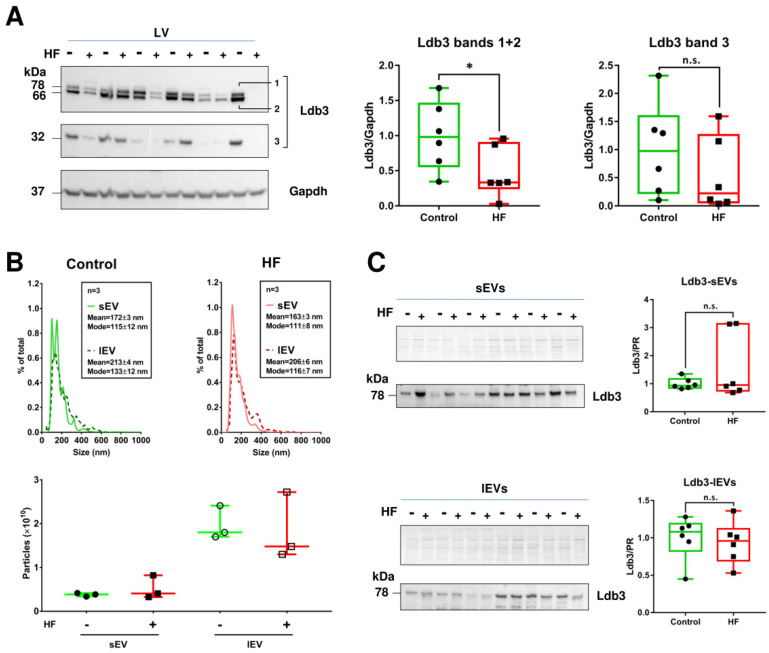
Ldb3 levels decrease in heart tissue, but not in cardiac EVs, during HF. (**A**) Western blots detecting Ldb3 expression in its 3 isoforms in left-ventricular tissue from control and HF patients. (**B**) NTA measuring particle size distribution (top panel) and concentration (bottom panel) of cardiac EVs isolated from the same patient tissues by UC. (**C**) Ldb3 detection in small (sEVs, top panel) and large EVs (lEVs, bottom panel). Data are expressed as median with interquartile range. * *p* < 0.05.

## Data Availability

The data presented in this study are available in “Full unedited gel for figures” Appendix A.

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
