# Peer review of "Lim Domain Binding 3 (Ldb3) Identified as a Potential Marker of Cardiac Extracellular Vesicles"

_ijms, 2022, doi:10.3390/ijms23137374_

Round 1

Reviewer 1 Report

In the paper named “Lim Domain Binding 3 (Ldb3) Identified as a Potential Marker of Cardiac Extracellular Vesicles” authors are looking for specific markers for the isolation and analysis of cardiac EVs in order to have a better understanding of their function in heart tissue. Therefore after a good characterization and validation authors suggest that Ldb3 can be a potential cardiomyocytes derived-EV marker and that this could be used to identify cardiac EVs in physiological and pathological conditions.

Only minor questions are required

1)      Some explanations of abbreviations were lost throughout the text such as HF, MI, LV ...

2)      In figure 1B author give information about 100K and 20K what means this?? In figure 1A there are 10000g and 2000g centrifugations. Please added to figure caption the information. In the same way in the text none about 100K 20K or 10000g or 2000g are referred however authors speak about small EVs and large EVs please clarify.

3)       The proteomic analysis is not clear. Firs author perform a MALDI-TOF analysis but they do not indicate the band used in this analysis. Are this analysis performed in gel from figure 1C? Why they perform this analysis if they have the possibility of make a shotgun analysis using LC-MS/MS? This technology gives a high number of protein identification and has a high accuracy. Author must clarify this part because a lot of information as an EV characterization is making using proteomics techniques.

4)      In figure 1 author says that perform a P values adjustment using a FDR how they make this?

5)      In point 2.2 author give information about the protein identified and how many proteins are related to cardiac function and how to plasma. Perhaps a venn diagram can clarify this part.

6)      Why author normalize the western blot using ponceau staining? Have author make some exosome markers that are expressed in most EV as CD63, Alix??

7)      In figure 2 caption authors say that they perform a 1D proteomic analysis what means this?

8)      The last paragraph in point 2.2 is confusing, please clarify

9)      How author explain that Ldb3 not appear in plasma western when it appear in plasma proteomics?

10)   How author are sure that the 3 bands found in Ldb3 western are protein isoforms and not posttraductional modification of this proteins produced in heart?

11)   Figure 5C explanation in the text are missing

Reviewer 2 Report

This is a very interesting article regarding the identification of a specific marker that could aid in the identification of cardiomyocite EVs. The article is in general well written, and the scientific methodology is sound. 

There are however some issues that have to be corrected for this article to be published. First, the introduction should be a little more detailed in explaining the importance of the thesis. A clear purpose/aim should be added in the last paragraph of the introduction.

Next, in the results section there are information that appertain to the materials and methods ection, starting with the first paragraph. Any type of information that shows how the study is performed should be added to the M&M section.

Regarding ethical approval - the authors state they have obtained IRB approval for the animal study, without any mention of the study on human biological products, which is mandatory according to the Declaration of Helsinki. Regarding informed consent - recent developments in the area or research ethics require a more specific type of consent than a purely general one, as was obtained, according to this article, by the authors. Also, as in the text is specified that IC was obtained, I do not understand why it was considered inapplicable in the declaration section of the manuscript.

The use of colloqvialism such as Interstingly should be avoided in a scientific manuscript.

Regarding the usefulness of Ldb3, according to the authors, is was higher in heart tissue, but present in other tissues as well. Also as its value decrease in ischemic heart disease, therefore potentially approaching brain levels? i feel that the conclusion of the study is not valid: " Our results suggest that Ldb3 is a potential cardiomyocytes derived-EV marker and could be used to identify cardiac EVs in physiological and pathological conditions."  and it should be reformulated to allow increased uncertainty regarding the results and their actual usefulness.
